# Light-Sheet Skew Ray-Based Microbubble Chemical Sensor for Pb^2+^ Measurements

**DOI:** 10.3390/s24216785

**Published:** 2024-10-22

**Authors:** Tingting Zhuang, Lukui Xu, Mamoona Khalid, Xuan Wu, Linqiao Du, Soroush Shahnia, Christophe A. Codemard, Zhiyong Bai, Ying Wang, Shen Liu, George Y. Chen, Yiping Wang

**Affiliations:** 1Shenzhen Key Laboratory of Ultrafast Laser Micro/Nano Manufacturing, Key Laboratory of Optoelectronic Devices and Systems of Ministry of Education/Guangdong Province, College of Physics and Optoelectronic Engineering, Shenzhen University, Shenzhen 518060, China; 2200453059@email.szu.edu.cn (T.Z.); 2100453032@szu.edu.cn (L.X.); dulinqiao@126.com (L.D.); baizhiyong@szu.edu.cn (Z.B.); yingwang@szu.edu.cn (Y.W.); shenliu@szu.edu.cn (S.L.); ypwang@szu.edu.cn (Y.W.); 2Shenzhen Key Laboratory of Photonic Devices and Sensing Systems for Internet of Things, Guangdong and Hong Kong Joint Research Centre for Optical Fiber Sensors, State Key Laboratory of Radio Frequency Heterogeneous Integration, Shenzhen University, Shenzhen 518060, China; 3Photonics and Communications Lab, Electrical Engineering Department, University of Engineering and Technology, Taxila 47050, Pakistan; mamoona.khalid@uettaxila.edu.pk; 4Future Industries Institute, UniSA STEM, University of South Australia, Mawson Lakes Campus, Adelaide, SA 5095, Australia; xuan.wu@unisa.edu.au; 5Laser Physics and Photonic Devices Laboratories, School of Engineering, University of South Australia, Mawson Lakes, SA 5095, Australia; soroush.shahnia@unisa.edu.au; 6TRUMPF Advanced Laser Laboratory, University of Southampton, Southampton SO17 1BJ, UK; christophe.codemard@trumpf.com

**Keywords:** fiber optic sensor, light sheet, skew rays, localized surface plasmon resonance, LSPR, lead sensor, Pb^2+^

## Abstract

A multimode fiber-based sensor is proposed and demonstrated for the detection of lead traces in contaminated water. The sensing mechanism involves using a light sheet to excite a specific group of skew rays that optimizes light absorption. The sensing region features an inline microbubble structure that funnels the skew rays into a tight ring, thereby intensifying the evanescent field. The sensitivity is further refined by incorporating gold nanoparticles, which amplify the evanescent field strength through localized surface plasmon resonance. The gold nanoparticles are functionalized with oxalic acid to improve specificity for lead ion detection. Experiment results demonstrated the significantly enhanced absorption sensitivity of the proposed sensing method for large center offsets, achieving a detection limit of 0.1305 ng/mL (the World Health Organization safety limit is 10 ng/mL) for concentrations ranging from 0.1 to 10 ng/mL.

## 1. Introduction

Safe drinking water is of vital importance to health and economic development. Contaminated drinking water can lead to various waterborne diseases, such as diarrhea and other infectious illnesses. Surface water and groundwater can become polluted by untreated sewage or chemical contaminants, including heavy metals, causing adverse health effects [1].

Lead (Pb) is a toxic heavy metal which poses a significant threat to both the environment and human health. It predominantly originates from industrial processes like mining, smelting, and manufacturing, as well as from its historical applications in products such as paint, fuel, and water pipes. One modern application of lead is in lead-cooled fast reactors (LFRs), which are advanced fourth generation nuclear reactor designs that employ liquid lead or lead–bismuth alloys as coolants instead of traditional light or heavy water [2]. Another main usage of lead is in the production of automotive lead–acid batteries. Improper waste management leads to lead ions (Pb^2+^) entering into the groundwater system and exposing humankind to adverse health risks, such as kidney disease, neurological damage, and suppressed immune system. To mitigate the problem of lead contamination in groundwater, the development of highly sensitive and reliable detection techniques is of paramount importance.

To detect Pb^2+^ in water, various techniques such as atomic absorption spectrometry, inductively coupled plasma spectrometry, and electrochemical methods have been explored [3,4]. The existing problems are that they are costly, time-consuming, and require a laboratory environment. Sensors based on localized surface plasmon resonance (LSPR) [5] fabricated on the surface of optical fibers have been widely demonstrated. LSPR leverages the unique optical response of metal nanoparticles [6,7], enabling the highly sensitive detection of chemical elements such as Pb^2+^. By using multimode fibers as the substate, the robustness and reusability of the sensor head is improved. However, there is still the need to further increase the sensitivity for detecting weak traces of lead as an early warning application.

We present a Pb^2+^ fiber optic sensor using a monolithic microbubble [8,9,10] as the sensing structure within multimode coreless fiber, which can provide both high sensitivity and robustness. This sensing structure features a hollow internal region encased by thin walls, in a similar way to a funnel. The light traveling through this thin circular waveguide generates a larger evanescent field, similar to an optical taper, consequently enhancing the absorption-based sensitivity. Gold nanoparticles have higher stability compared to silver nanoparticles [11,12,13]. Incorporating gold nanoparticles (Au NPs) into multimode coreless fiber facilitates the enhanced sensitivity due to plasmonic resonance from gold NPs, which strengthens the interaction between light and matter, thereby increasing the light absorption efficiency and the sensitivity.

Separate comparisons of the three enhancement methods have previously been published, which has accumulated step-by-step up to this work. 1. (More reflections [14]) Compared skew rays to other excitation methods using a uniform structure. 2. (Absorption enhancement [15]) Compared LSPR and no-LSPR using skew rays and a uniform structure. 3. (Larger evanescent field [16]) Compared microbubble vs. uniform structure using skew rays. Since these three enhancement effects belong to complementary mechanisms, they can stack without conflict (e.g., more chances for light to interact with matter, stronger absorption per interaction, and more light to interact with matter). The main difference in this work is the combination of microbubble and LSPR; surface functionalization for lead (II) sensing; the optimization of the skew rays input conditions for the microbubble; and the characterization of the Pb^2+^ sensing metrics.

## 2. Materials and Methods

### Fiber Structure Fabrication

A 3 cm long sensing fiber was prepared by removing its cladding. The sensing fiber uses a coreless multimode fiber with a cladding diameter of 320 µm and a coating layer diameter of 483 µm. At a 532 nm wavelength, the refractive index of the cladding layer is roughly 1.46 and the refraction of the coating layer is 1.38. The fiber was precisely cleaved into two parts with flat ends using a fiber cleaver (Fujikura CT104+, Japan). A femtosecond laser-based micromachining system (Newport FemtoFBG, Irvine, CA, USA) was utilized to produce a square microcavity of ~100 µm and a depth of ~20 µm onto the surface of the fiber. The femtosecond laser (FSL) for the micromachining process operated at a central wavelength of ~800 nm and can produce a spot size of 1.02 µm using a 20× microscope objective with a numerical aperture (NA) of 0.5. The pulse width of the FSL was 290 fs, its repetition rate was set to 10 kHz, and its pulse energy was set to 0.15 μJ (~1.5 mW). The fiber sample for micromachining was moved on a 3-axis translation stage with a speed of 5 mm/s.

After the micromachining process, the sensing fiber surfaces were cleaned with ethanol to remove debris. The two end surfaces were then aligned and fused together using a fusion splicer (Fujikura FSM-100P+, Japan), which applied heat through an electric arc to create the splice. This thermal process resulted in the formation of a micrometer-scale inner air bubble within the optical fiber. Since the sensing fiber was constructed from two fiber sections spliced together, and since a halo pattern can be observed at the output of the fiber, it is evident that a portion of skew rays are maintained even after propagating through the splice point and the microbubble region. The microbubble parameters are governed by the micromachining and splicing parameters, contributing to the sensor’s overall functionality and performance.

It should be noted that the fabricated microbubble is imperfect due to the presence of a smaller air bubble adjacent to the main microbubble, as shown in Figure 1. The exact state of the microbubble could not be fully controlled during the thermal fusion splicing process, possibly because the microcavity inscribed on the end face of the fiber was too large. A smaller microcavity allowed for the formation of a more circular microbubble structure. However, thinning a section of the wall by introducing a side microbubble increased the evanescent field to some extent. Since the microbubble is formed within the original fiber, the microbubble structure is monolithic, and thus coupling loss is minimal.

The exposed part of the optical fiber was flushed with piranha solution (1:3 (*v*/*v*) H_2_O_2_ and H_2_SO_4_), then rinsed multiple times with deionized water. Following this treatment, the fiber was immersed in poly (diallyl dimethylammonium chloride) (PDDA) solution for a soak time of 20 min. It was then rinsed again with deionized water and allowed to dry in an ambient environment. The single layer of PDDA is an ideal intermediate layer for minimizing the separation distance between nanoparticles and the fiber interface to facilitate efficient evanescent wave-based interactions.

Au NPs with an average diameter of approximately 20 nm were synthesized utilizing a sodium citrate reduction method. In order to align the resonant wavelength of the gold nanoparticles associated with LSPR to the optical wavelength of the laser and the peak absorption wavelength of Pb^2+^, the size of the gold nanoparticles was chosen to be ~20 nm on average. Size shape and aggregation effects affect the resonance wavelength, although size has less of an impact in the case of spherical nanoparticles; the commercial availability and cost-effectiveness were also considered. A 49.5 mL volume of water was boiled with a magnetic stirrer in place. Furthermore, 0.5 mL of chloroauric acid aqueous solution was introduced in water at the boiling point of water. Then, 1% sodium citrate solution was prepared by dissolving 0.1 g of sodium citrate in 10 mL of water. Additionally, 2 mL of this sodium citrate solution was then added to the boiling water and stirred until a color change was observed. The solution was allowed to boil for an additional 6 min until the solution turned wine red. The peak absorbance wavelength of the nanoparticles suspended in water, measured using a UV–visible spectrophotometer (METASH UV-8000, Shanghai, China), was found to be approximately 521.7 nm. The peak absorption wavelength (537.8 nm) of Au NPs attached to the optical fiber surface was characterized by launching white light into the optical fiber and measuring the transmittance with an optical spectrum analyzer (OceanOptics QE65000, Shanghai, China). The absorption band for Pb^2+^ ranges from the ultraviolet to the visible regime. Lead nitrate, for example, has peak absorption wavelengths concentrated in the ultraviolet band. The Au NPs solution was then mixed with an oxalic acid solution and allowed to react for 1 h. Following this, the chosen sensing fiber section was immersed in the mixture of the Au NPs and oxalic acid solutions, repeating this immersion process five times with 5 min intervals between each iteration.

## 3. Results

### 3.1. Principles of Skew Rays

Higher order modes propagating in optical fibers have a greater evanescent field distribution [17] beyond the core–cladding interface compared to the fundamental mode [18]. This phenomenon results in deeper light penetration and enhanced interaction with materials on the fiber surface and subsequently improves the sensitivity of evanescent wave-based sensors [19]. Since highly multimode fiber is used, from the ray optics perspective, when light enters the fiber from the end face, it does not propagate strictly along the vertical axis of the cross section (meridian direction); instead, it forms an angle and is thus referred to as a skew ray [20,21,22]. The resulting skew ray is described mathematically in terms of two angles relative to the waveguide–external interface, as illustrated in Figure 2.

For a given angle of incidence θ, skewed rays undergo significantly more reflections during propagation compared to radial rays [23], with the number of reflections increasing by several orders of magnitude relative to meridional rays. This phenomenon can extend the overall penetration path of light within the surrounding medium, thereby enhancing sensitivity to variations in the external environment [24,25].

The proposed enhancement mechanism employs localized surface plasmon resonance (LSPR). As illustrated in Figure 3, the Au NPs exhibit extremely small dimensions, much smaller than the incident wavelength, in the sensing region of the optical fiber. When exposed to light, the conduction electrons within the nanoparticles oscillate in phase, leading to the polarization of charges on the particle surface. These charges resonate with the incident light at a specific frequency, influenced by a restoring force [26,27]. Oxalic acid is utilized to functionalize the Au NPs in this process. The molecule contains two hydroxyl groups (-COOH): one binds to the Au NP, while the other detects Pb^2+^ ions due to the strong affinity between hydroxyl groups and Pb^2+^ ions. Compared to other metal ions, oxalic acid has a higher specificity for lead ions [28]. To mitigate the reduction in light–matter interaction sites caused by the low concentration of Au NPs [26], a layer of PDDA is applied to achieve electrostatic adsorption. In general, contaminants can increase the expected signal due to cross sensitivity or decrease it due to a blocked sensing region.

### 3.2. Experiment Setup

A fully automated microchannel platform with six computer-controlled pumps (XFP01-BD, Xunfei Scientific Instruments Co., LTD, Suzhou, China) was built to excite the skew rays and LSPR modes, as shown in Figure 4. This platform facilitates highly efficient and reliable measurements. Among them, two pumps use air and pure water for rinsing and drying the optical fiber after each measurement in order to ensure reliable measurements. The remaining four pumps are used for delivering four different concentrations of the analyte under test. All measurements were performed using a multimode coreless fiber made of fused silica with a rod diameter of 320 µm, a coating diameter of 400 µm, a numerical aperture of 0.4, and a total length of 105 cm. To probe the sensing fiber, a non-polarized single-mode laser source (Thorlabs DJ532-10, LDM56DJ, Newton, NJ, USA) with a wavelength of 532 nm was used. The probe light had a beam diameter of 3.2 mm and was collimated through a plano-convex lens with a focal length of 150 mm. The collimated beam was converted into a light sheet using a cylindrical plano-convex lens with a focal length of 100 mm (the input light sheet thickness was about 20 µm). A portion of the collimated light is tapped and used as a reference for laser intensity drift correction, as detected by a photodiode (PDA60A2, Guangyi intelligent technology Co., LTD, Guilin, China). The input end of the sensing fiber is centered and fixed on a computer-controlled rotation stage (Thorlabs PRMTZ8, Newton, NJ, USA). The fiber alignment relative to the incoming beam can be fine-tuned by a three-axis translation stage. The sensitivity can be optimized by finding the optimum effective incidence angle of light, which depends on two angles in the case of skew rays. Hence, both angles are tuned in a systematic matter via the center offset (subsequently denoted by “height”) and launch angle (horizontal relative to the normal) θ in order to realize the widest possible range of effective angles and compare the corresponding absorption.

The output light from the sensing fiber is fully captured by an integrating sphere power meter (Newport 918D-IS-1,1936-R, Irvine, CA, USA). Dilutions of lead ion solution are injected into the microfluidic tubing by an automated microfluidic syringe pump using a flow rate of 1 mL/min. To ensure the accuracy of each concentration, the microchannel is rinsed with pure water (pumped) and dried with air (pumped) after each measurement. The switching valve (LSG-1D-6T-A6, Xunfei Scientific Instruments Co., LTD, Suzhou, China) can be aligned to different channels by computerized control. To avoid instability and measurement inaccuracies, it is important to avoid the contamination of the sensing fiber by liquid residue, such as from finger contact or human breath.

After the fiber input end face is positioned at the center of rotation on the motorized stage and the optical fiber straightened, the axis of the fiber can be located by scanning the launch angle of light and the fiber clamp height. At a zero incidence angle (fiber axis), the received optical power should be maximum and should be at the peak value of the symmetrical curve of angle vs. power (2D graph) or the peak value of a horizontal line of angle vs. height vs. power (3D graph).

Note that for practicality and portability, the sensing system can be simplified using a compact and low-cost light source and a series of lenses, and in fixing the optical fiber at the pre-calibrated optimum angle/position, a simplified non-moving system can be obtained. In addition, the integrating sphere can be replaced by a large-area photodiode within an enclosed package.

### 3.3. Sensitivity/Detection Limit Characterization

The sensor characterization covered Pb^2+^ concentrations from 0.1 ng/mL to 100 ng/mL, with each concentration level differing by one order of magnitude; a total of four concentrations of lead ion solutions were measured. The experimental results are based on power attenuation as a metric of sensing performance, which shows the responsiveness of the sensor to changes in the analyte concentration. The residual probe power at the output of the optical fiber was measured in each case and was compared to deionized water as a baseline. The output power for each concentration of solution was calculated using the output power under pure water conditions as a base, and the resulting value was used as the power attenuation.

Figure 5 shows the attenuation of probe light for different concentrations of lead ion solutions for different combinations of θ and fiber center offsets. The attenuation peaks are located at higher center offsets, indicating fiber edge excitation. For a concentration of 10 ng/mL, the peak attenuation of 4.96 dB occurs at θ = 31.0 deg and a central offset of 160 µm. This implies that most of the skew rays propagate in the coating, leaving only a small portion of light to be guided in the optical fiber. High sensitivity is usually produced under extreme angular conditions when light attenuation is severe due to a large number of total internal reflections. At higher concentrations, the attenuation hot spot gradually diminishes to zero. This phenomenon is likely attributable to the binding of lead ions to the Au NPs, which consequently detunes the resonant wavelength of the Au NPs. LSPR is generally independent of the angle. However, a correlation can be observed when a uniform layer is present between the waveguide and Au NPs, leading to varying attenuation levels at different incident angles [29].

The relationship between the concentration of the lead ion solution and the probe light attenuation was investigated in the range of 0.1 to 100 ng/mL based on the three nominal positions of (i) θ = 31.0 deg, 160 µm center offset, (ii) θ = 34.0 deg, 160 µm center offset, and (iii) θ = 37.0 deg, 160 µm center offset. Within this range of concentrations, the change in the refractive index in the ambient medium or near the nanoparticles can lead to a redshift of the absorption band and a change in the absorption intensity [30]. The resonance conditions are very sensitive to the environment surrounding the nanoparticle or the thin dielectric layer present on the particle, thus adsorption on the Au NPs can lead to a redshift of the LSPR band. Similar to most fiber optic chemical sensors, the sensitivity or the relationship between the measurand and the detected parameter is usually nonlinear [31,32,33]. As shown in Figure 6, the operating range is from 0.1 ng/mL to 10 ng/mL, and the attenuation of the sensor is approximately linearly proportional to the concentration. According to the linearly fitted gradient, the uniform sensitivity is 0.0138 dB/dB _(re. 0.1 ng/mL)_ between 0.1 and 10 ng/mL at θ = 31.0 deg with a center offset of 160 µm; at θ = 34.0 deg with a center offset of 160 µm, it is 0.0774 dB/dB _(re. 0.1 ng/mL)_; and at θ = 37.0 deg and a center offset of 160 µm, the sensitivity is 0.0717 dB/dB _(re. 0.1 ng/mL)_.

The maximum sensitivity is 0.0974 dB/dB _(re. 0.1 ng/mL)_ under the launch conditions of (160 µm, 34°). The standard deviation of noise was measured to be 0.0682 dB, which resulted in a limit of detection LoD [34] of 0.1305 ng/mL. Around the LoD concentration, the signal strength is comparable with that of noise, which is still detectable but with less confidence than that of higher concentrations. Hence, the lowest concentration tested was 0.1 ng/mL, which is slightly lower than the LoD of 0.1305 ng/mL. To lower the concentration further would result in considerably reduced reliability. The length of the sensing region was 3.2 cm, and the length-normalized attenuation was 1.55 dB/cm (10 ng/mL) under emission conditions (160 µm, 31°).

To examine the transient response, which is useful for real-time measurements involving changing analyte concentrations, the variation in the solution concentration over a 30 min period was analyzed under a sampling interval of 2 s. It is not uncommon to cycle between two or three concentration levels to observe the dynamic behavior of the system [35]. Figure 7 illustrates the changes in residual probe power at the power meter as the concentration shifts from low to high and subsequently from high to low. The low concentration denotes pure water, and the highest concentration represents a lead ion concentration of 1 ng/mL. As the concentration increased, a corresponding decrease in transmitted probe power was observed. The observed power fluctuations are attributed to ambient air currents affecting the free-space light propagation prior to entering the optical fiber and the presence of microscopic air bubbles within the lead ion solution, which can cause unpredictable scattering and attenuation.

## 4. Discussion

In an optimal scenario, larger microbubbles create a thin-wall structure at the microbubble region, funneling light into a narrow corridor similar to the effect of an optical taper. Depending on the angle of incidence, this phenomenon can enhance the attenuation field and can potentially increase the number of total internal reflections as well as the evanescent field. Consequently, this increases the sensitivity to variations in the external environment. This sensitivity is further amplified by the LSPR effect. However, the fabrication of large microbubble structures was not viable due to the limitations of the fabrication process. The enhancement in sensitivity compared to a homogeneous fiber [16] is mainly attributed to the increase in the refractive index around the microbubbles, caused by the expansion of air and compression of silica. This should redshift the LSPR resonance wavelength toward the laser wavelength, thus increasing the absorption of light.

Optical sensors based on skew rays utilize a uniform distribution of light–matter interactions and exhibit high sensitivity, high optical stability, temperature insensitivity, and high robustness. Intensity-based measurements are widely known to be more stable and less sensitive to temperature drifts compared to phase-based measurements (e.g., interferometry). Additionally, there is a lower requirement for the laser source in terms of polarization and linewidth. Previous works on multimode fibers excited with skew rays have already demonstrated insignificant temperature cross sensitivity [36], even when using polymer fibers where the thermo-optic coefficient of PMMA (–1 × 10^−4^/°C) is larger than that of silica (8 × 10^−6^/°C). As for robustness, large-diameter multimode fibers are inherently sturdier than, for example, fragile nanofibers.

The higher loss associated with higher order modes in large-diameter coreless fiber limit the application to short-range energy transmission, which is not suitable for information transmission. However, high-loss fibers can still be used for point sensors, as shown by the many examples in the literature that use high-loss polymer fibers [37] and nanofibers [38] for highly sensitive short-range measurements.

Overall, the proposed method in this work achieved a relatively small LoD while retaining a robust sensor head that can be reused. In Table 1, the performance of different lead (II) sensors is compared.

A microbubble structure with a larger diameter is required to increase the sensitivity and thus improve the detection limit of the sensing system. Employing a collimated light sheet rather than a collimated beam enhances the interaction between the light and the external environment, thereby amplifying the sensitivity. This configuration minimizes the instability in power absorption that is typically associated with transverse mode competition, as the narrower group of rays mitigates these effects [45]. Therefore, the experimental setup needs to be optimized to improve the stability of the system. This can be accomplished by employing cylindrical lenses with enhanced precision to reduce the thickness and divergence angle of the light sheet. Specifically, lenses that exhibit minimal spherical aberration are suitable for achieving this level of refinement. The Au NPs attached to the fiber surface are susceptible to dislodgement by liquid flow within the microfluidic channels, potentially leading to their detachment. For further optimization, the femtosecond laser inscription of nanocavities is an optimization strategy to improve the adhesion strength of Au NPs.

## 5. Conclusions

We demonstrated a new type of lead (II) sensor using an fs laser micro-machined microbubble inline structure paired with LSPR, interrogated by light sheet-excited skew rays. This sensor comprises several stages of enhancement: (1) skew rays increase the interaction strength between light and matter; (2) a light sheet ensures the precise excitation of a narrow group of skew rays; (3) the microbubble structure can achieve higher sensitivity due to the funneling of light into a tight ring within its walls; (4) LSPR generated by the adsorbed Au NPs further enhances the evanescent field; and (5) the oxalic acid surface functionalization helps to specifically detect lead ions. Through the systematic testing of the launch conditions of light to optimize performance, a sensitivity of 0.0774 dB/dB _(re. 0.1 ng/mL)_ and a detection limit of 0.1305 ng/mL were observed. The main application is the detection of lead ion concentrations in contaminated water, which can contribute to improvements in global health.

## Figures and Tables

**Figure 1 sensors-24-06785-f001:**
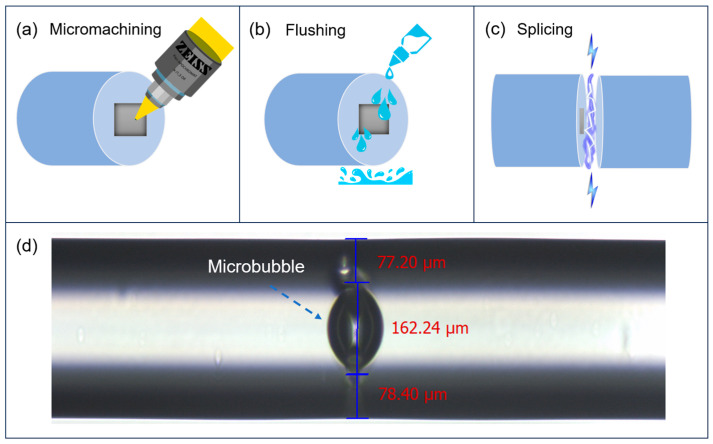
Microbubble fabrication process. (**a**) Femtosecond laser micromachining of a microcavity. (**b**) Ethanol cleaning of fiber end face. (**c**) Microbubble formation from discharge heating of fusion splicer. (**d**) Microscope image of microbubble structure within multimode coreless fiber.

**Figure 2 sensors-24-06785-f002:**
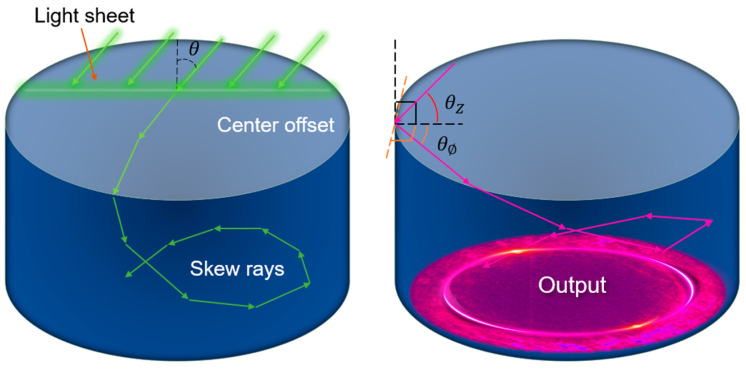
Illustration of skew ray generation and propagation.

**Figure 3 sensors-24-06785-f003:**
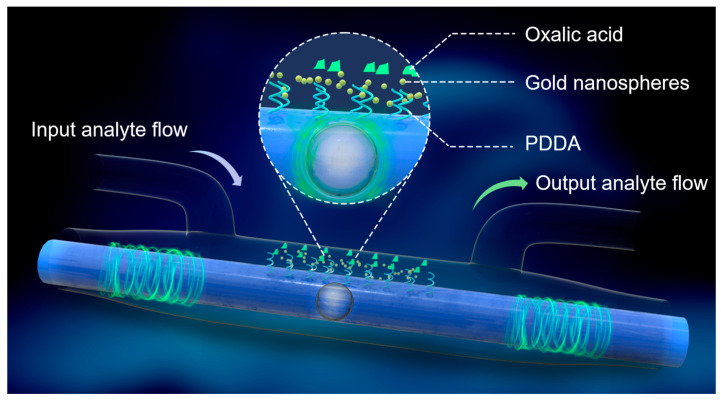
Schematic of the microbubble-based fiber optic Pb^2+^ sensor.

**Figure 4 sensors-24-06785-f004:**
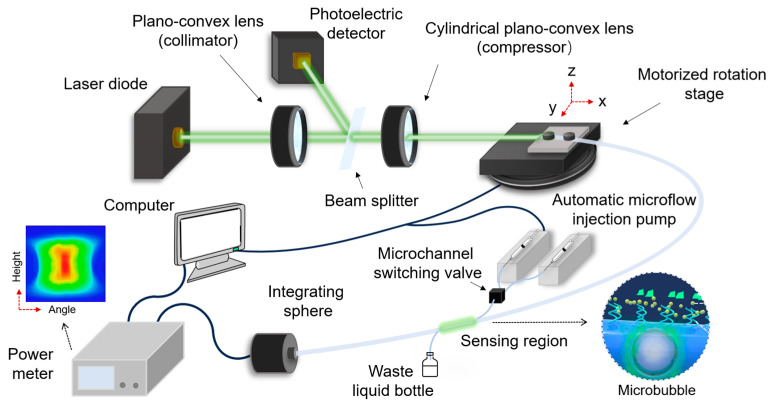
Experimental setup to generate light-sheet skew rays and localized surface plasmon resonance for Pb^2+^ sensing.

**Figure 5 sensors-24-06785-f005:**
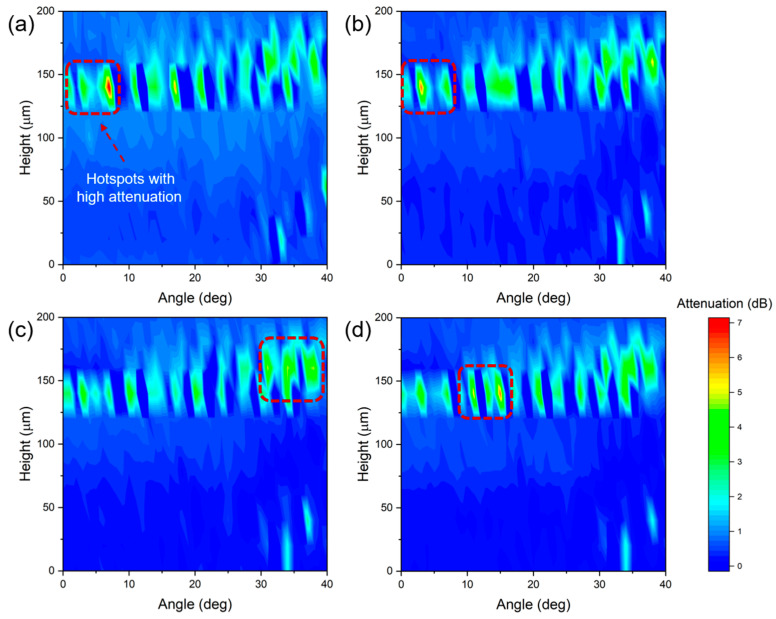
Attenuation (color depth) induced by Pb^2+^ relative to pure water as a function of the launch angle and center offset. (**a**) 0.1 ng/mL; (**b**) 1 ng/mL; (**c**) 10 ng/mL; (**d**) 100 ng/mL.

**Figure 6 sensors-24-06785-f006:**
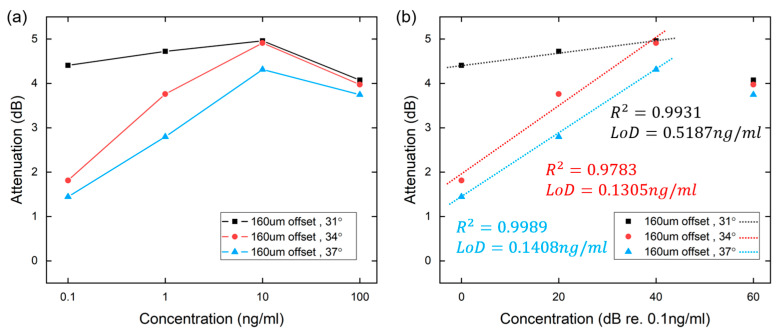
Relationship between concentration and attenuation. (**a**) Logarithmic *x*-axis. (**b**) Linearized *x*-axis (dB relative to 0.1 ng/mL).

**Figure 7 sensors-24-06785-f007:**
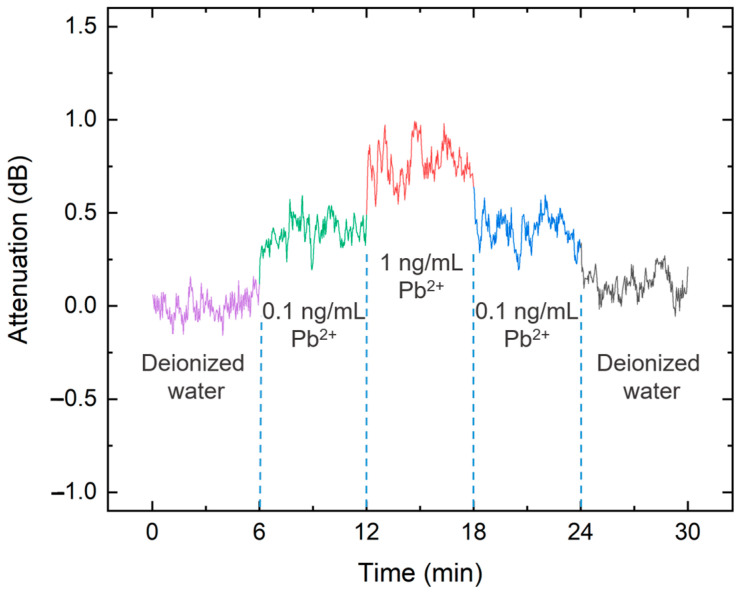
Transient response to changes in lead ion concentration between 0 and 1 ng/mL.

**Table 1 sensors-24-06785-t001:** Comparison of analytical performance between our light-sheet skew ray-based sensors and previously reported sensors for Pb^2+^ detection.

Method	Functional Material	Linear Range	LoD	Reference
Integrated Michelson interferometer	MXene	~	0.286 ppb	[39]
Colorimetric and fluorescent detection	Galloyl-PDA	0–10 µM	1.329 µM	[40]
Hybrid fiber (LPG + FBG) grating	CCS-NGO/PAA	~	0.5 nM	[41]
PCF surface plasmon resonance interferometer	Gold layer	0–100 ppm	8.32 ppm	[42]
PCF modal interferometer	Chitosan-PVA, GSH functionalized AuNPs	~	1.6 ppb	[43]
Film-enhanced microfiber interferometer	PDA-MA	2 × 10^−7^–2.072 × 10^−2^ ppb	0.1678 ppb	[44]
Light-sheet skew rays with inline microbubble	Au NPs-Oxalic acid	0.1–10 ppb	0.1305 ppb	This work

## Data Availability

Data underlying the results presented in this paper are not publicly available due to privacy but may be obtained from the authors upon reasonable request.

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
