# Peer review of "Light-Sheet Skew Ray-Based Microbubble Chemical Sensor for Pb2+ Measurements"

_sensors, 2024, doi:10.3390/s24216785_

Round 1

Reviewer 1 Report (New Reviewer)

Comments and Suggestions for Authors

This paper presents an MMF device in which the sensitivity to the concentration of Pb2+ in water is enhanced by a microbubble and gold NPs deposition on the sensitive section. I have the following comments:

1. I am not sure that amplification of the evanescent field is the correct term. I see an increased sensitivity due to plasmonic resonance from gold NPs, so enhanced may be better.

2. In the 5th paragraph of section 2.1 the authors wrote:

      ... Since nanoparticle size do not dramatically affect the resonant wavelength...

Size shape and aggregation effects affect the resonance wavelength, although size affects less in the case of spherical nanoparticles. 

3. How was the modified absorption spectrum measured with the NPs deposited on the fiber surface? It was measured with the NPs deposited on a glass-slide?

4. In real conditions, contaminated water may contain additional contaminants (other metals or chemicals). How may these conditions affect the sensor performance? 

Author Response

Reviewer 2 Report (New Reviewer)

Comments and Suggestions for Authors

First of all, it is not entirely clear why only four concentration values were used in the measurements and how then was determined the range of these values for the proposed sensor. Moreover, in some experiments were used only two values of concentration (e.g. fig. 7). It would be interesting to know how the sensor behaves at the other concentrations inside and outside the specified range.

Further, the determined LoD is more than 0.1 ng/mL. What does it mean? Doesn't that mean the sensor cannot work properly at that concentration?

And finally, the principle according to which the parameters of the setup (center offset and angle) were chosen for determination of maximum sensitivity is not entirely clear. In my opinion, the maximum sensitivity will observed at the points where there is very sharp transition between colors in fig. 5, e.g. near 9 deg and 11 deg.

Comments on the Quality of English Language

I am not a native English speaker, that is why all my comments are advisory in nature. There are some typos (e.g. word "substrate" in str. 61) that need to be corrected. Besides, there are some expressions the use of which seems questionable to me. The words "pump power", "pump light" I would change to "probe light" may be. I do not quite understand the phrase "attenuation field" (str. 284). Perhaps it was meant "attenuation of the field"? In addition, the use of word "viable" in str. 288 seems strange to me.

Author Response

This manuscript is a resubmission of an earlier submission. The following is a list of the peer review reports and author responses from that submission.

Round 1

Reviewer 1 Report

Comments and Suggestions for Authors

The authors demonstrated a fiber-optic probe for lead(II) detection, using a combination of various methods including microbubble, skew rays and local surface plasmon resonance. This combined method is new and the ability to quantify lead(II) at about a tenth of the World Health Organization proposed limit makes it feasible for trace detection of lead contamination. I recommend its publication if the authors can address the following comments/suggestions:

1. How do the authors ensure each concentration measurement is accurate, and does a lack of rigorous rinsing affect measurements in a real environment outside the lab?

2. A comparison with other lead(II) sensors based on optical fiber is missing and should be added.

3. Can the authors describe in greater detail of how the entire system can be made compact and still retain optimum light-launching conditions?

4. Suggest adding text labels to the illustrated procedure in Figure 1 for better readability. Also, the microbubble should be labelled.

5. Figure 4 contains an abbreviation PD which should be replaced by its full name.

6. The formatting of the references needs to be consistent.

Reviewer 2 Report

Comments and Suggestions for Authors

Please refer to the attachment for details

Comments on the Quality of English Language

English is good

Reviewer 3 Report

Comments and Suggestions for Authors

The presented article is devoted to the recently quite interesting issue of using the LSPR phenomenon to detect lead ions in contaminated water, and the authors' use of the original skew beam technique allows achieving fairly low limit concentrations. The article is quite well written, in clear, understandable language, but there are some comments (or suggestions) for its improvement:

1. The authors refer to their previous works too much out of 30 references, 8 to themselves (in the introduction - references 8-12) does no one in the world work with such systems

(in part 3.1 Principles of skew rays - references 15-18)

2. In the introduction, it would be good to justify in more detail the use of gold nanoparticles, because it is known that silver nanoparticles have better properties for LSPR.

3. It would also be good to justify the size of the gold nanoparticles used, because the properties of spr will change, although not to a critical extent, depending on the size.

4. When describing the Fiber structure, the authors use the notation um, while in the figures (μm) and further in the text, μJ are used. It is necessary to convert to the generally accepted notation μ.

5. In reference 10, the list of authors is given twice.

Overall, the work is very good and can be published in the journal Sensors after making corrections.

Round 2

Reviewer 2 Report

Comments and Suggestions for Authors

see attachment

Comments on the Quality of English Language

Minor editing of English language required.